# Structural and Functional Differences in Small Intestinal and Fecal Microbiota: 16S rRNA Gene Investigation in Rats

**DOI:** 10.3390/microorganisms12091764

**Published:** 2024-08-25

**Authors:** Xiao-Wei Sun, Hong-Rui Li, Xiao-Lei Jin, Xiao Tang, Da-Wen Wang, Xiao Zhang, Jian-Gang Zhang

**Affiliations:** Pathology Institute, School of Basic Medical Sciences, Lanzhou University, Lanzhou 730000, China

**Keywords:** ileum, feces, microbiota, 16S rRNA gene, sequencing, rats

## Abstract

To compare the differences in floral composition and functions between the two types of microbiota, ileal contents and feces were collected from Sprague Dawley (SD) rats fed in a conventional or specific-pathogen free (SPF) environment and rats fed a high-fat diet (HFD), and the V3–V4 region of the 16S ribosomal ribonucleic acid (rRNA) gene in these rats was then amplified and sequenced. Compared with feces, about 60% of the bacterial genera in the ileum were exclusive, with low abundance (operational taxonomic units (OTUs) < 1000). Of bacteria shared between the ileum and feces, a few genera were highly abundant (dominant), whereas most had low abundance (less dominant). The dominant bacteria differed between the ileum and feces. Ileal bacteria showed greater β-diversity, and the distance between in-group samples was nearer than that between paired ileum–feces samples. Moreover, the ileum shared various biomarkers and functions with feces (*p* < 0.05). The HFD and SPF conditions had a profound influence on α-diversity and abundance but not on the exclusive/shared features or β-diversity of samples. The present findings suggested that, under conventional circumstances, fecal bacteria can represent approximately 40% of the low abundant ileal bacterial genera and that dominant fecal bacteria failed to represent the ileal dominant flora. Moreover, fecal flora diversity does not reflect β-diversity in the ileum.

## 1. Introduction

Trillions of bacteria exist in the human gastrointestinal (GI) tract [1]. With the help of the high-throughput sequencing of nucleotides and the identification of the 16S ribosomal ribonucleic acid (rRNA) gene as highly conserved in most bacterial clades, [2], researchers have been able to identify and classify gut microbiota, which are distributed throughout the entire GI tract in diverse microbial communities [1]. Gut bacteria have been shown to take part in many physiological or pathological processes in humans, such as inflammatory bowel disease, cancer, nutritional obesity, diabetes, and cardiovascular disease [3,4,5,6,7,8,9,10,11]. However, based on the idea that the colon is the site of final GI content collection and the fact that colonic samples are easily acquired, most researchers have obtained samples from feces and regarded fecal flora as being naturally synonymous with gut microbiota [12,13,14,15,16].

During transportation and digestion of food components through the gut, different biochemical inner environments are formed in the small intestine (SI) and large intestine (LI). Accordingly, it is highly possible that the microbiota of these two organs are distinct [15]. The most significant distinction between SI and LI microbiota might be the persistent clinical fact that the SI has a remarkably different incidence of various diseases, such as carcinoma, from LI. Therefore, identifying differences and relationships between SI and LI microbiota is crucial in clarifying the gut bacterial mechanisms of various disorders. Several clinical and laboratory animal observations have proven the marked differences between SI and stool microbiota, as well as between different segments of the SI [17,18]. A detailed microbiome profile comparing the ileum and feces, however, is still lacking. To address this, we collected the ileal contents and feces of rats and compared their structural and functional features based on 16S rRNA gene sequencing and bioinformatic analysis.

## 2. Materials and Methods

### 2.1. Animals

Male Sprague Dawley (SD) rats, ages 4–5 weeks and weighing 52.32 ± 8.48 g, were raised individually in a conventional setting for 3 months, with diet and water ad libitum, to simulate the natural state (ordinary rats, n = 6). The rats’ chow (3.16 kcal/g) was produced by KeAo-XieLi Co., Ltd. (Beijing, China) and sterilized with cobalt-60 (^60^Co). In addition to the regular rats, another two sets of rats were also used for detection. The first set of rats (n = 10) were fed a high-fat diet (HFD) for 60 days and then switched to chow for another 60 days to identify the long-term impact of HFD on gut microbiota composition (HFD rats). The HFD (5.58 kcal/g with 67% energy intake from fat; 16% protein, 42% carbohydrates, and 31% fat) was produced using chow, egg yolk powder, lard, and sugar [19]. Another set of clean rats with a body weight of 304.10 ± 29.92 g that were raised in a specific-pathogen–free (SPF) environment and fed a chow diet and water ad libitum were chosen randomly (clean rats, n = 6) (Figure 1). All methods were carried out in accordance with the relevant guidelines and regulations of the Laboratory Animal—Guidelines for Ethical Review of Animal Welfare (GB/T 35892–2018) (access on 1 May 2019), and all experimental protocols were approved by the Medical Ethics Committee of Lanzhou University (jcyxy20190302). All methods were reported in accordance with the ARRIVE guidelines (https://arriveguidelines.org, access on 1 May 2019) for the reporting of animal experiments.

### 2.2. Sample Collection

Rats were anesthetized by intraperitoneal injection of pentobarbital sodium (5 mg/100 g body mass) and killed by portal-vein blood drainage. We expressed no less than 2.0 g content of 5.0–7.0 cm of ileum or colon into sterile Eppendorf tubes (Eppendorf, Hamburg, Germany). Samples were then stored in a −80 °C refrigerator after quick freezing in liquid nitrogen. To avoid environmental contamination, we performed our experiments on a clean bench.

### 2.3. DNA Extraction, PCR, and Sequencing

We extracted total deoxyribonucleic acid (DNA) from 0.5 g of each sample using a magnetic soil and stool DNA kit (#DP812; TIANGEN Biotech Co., Ltd., Shanghai, China) per the kit protocol. Primers were designed according to the conserved V3–V4 region of the 16S rRNA gene. The sequences were as follows: 338F: 5′-ACTCCTACGGGAGGCAGCA-3′; 806R: 5′-GGACTACHVGGGTWTCTAAT-3′ [20]. The target region was amplified using a polymerase chain reaction (PCR) with the barcode at the end of the primer. The amplification procedure was as follows: 95 °C for 5 min, with 25 cycles of 95 °C for 30 s, 50 °C for 30 s, and 72 °C for 40 s, followed by an extension at 72 °C for 7 min. The products were then purified, quantified, and homogenized to form a sequencing library and run on a 1.8% agarose gel. After quality inspection and preparation of a flow cell chip, we sequenced 500 ng of PCR products by PE 250 mode (2 × 250 bp paired ends), performed on an Illumina HiSeq 2500 platform (Illumina, Inc., San Diego, CA, USA; via BMK Biotechnology Co., Beijing, China). The sequencing length was 350–450 bp. We transformed original image data files into sequenced reads via base calling. DNA extraction, PCR, gene sequencing of the V3–V4 region of the 16S rRNA, data preprocessing, and outcome assessment were conducted by BMK Biotechnology Co. (Beijing, China), accessed on 28 February 2020 and 25 November 2020, BMK191202–X098–01 and BMK200916–AC763–0101), who had no vested interest in the experiment and were not aware of the group allocation.

### 2.4. Data Preprocessing

According to overlap relationships, we spliced PE reads using Fast Length Adjustment of SHort reads (FLASH) software v1.2.11 (Johns Hopkins University Center for Computational Biology, Baltimore, MD, USA) [21]. Raw tags were filtered for quality using Trimmomatic software (v0.33, GitHub Inc., San Francisco, CA, USA) (https://USADELLAB.org) [22]. We removed chimeras using UCHIME software v4.2 [23] to form clean tags, which were then clustered into operational taxonomic units (OTUs) at 97% sequence similarity using USEARCH software (v10.0) (https://www.drive5.com/usearch/) [24]. Representative sequences of OTUs were compared with the Silva microbial reference database v128 (http://www.arb-silva.de) [25], and OTUs were annotated using RDP Classifier (v2.2, Michigan State University, East Lansing, MI, USA) (https://rdp.cme.msu.edu) [26]. We generated species richness at different taxonomic levels using Quantitative Insights into Microbial Ecology (QIIME) software (v2.2, http://qiime.org/). Alpha-diversity indices and community dissimilarity (β-diversity) were evaluated using QIIME2 (https://qiime2.org/) and R software v3.1.1 (picante package v1.8.2 and vegan package v2.3-0, respectively) (R Foundation for Statistical Computing, Vienna, Austria). Software and websites were accessed on 28 February 2020 and 25 November 2020.

### 2.5. Statistics

Data were expressed as mean ± standard deviation (SD) or standard error of the mean (SEM). We conducted a *t*-test using SPSS v22 (IBM Corp., Armonk, NY, USA) for comparisons of the alpha diversity indices between ileum and feces. Principal component analysis (PCA) and principal coordinate analysis (PCoA) were analyzed by the binary Jaccard method, and the community dissimilarities were analyzed via permutational multivariate analysis of variance (PERMANOVA). The predicted functional profiles between ileal and fecal microbiota were analyzed using a *t*-test. Differences were considered statistically significant when *p* < 0.05.

## 3. Results

### 3.1. Large Proportions of Ileal Flora Were Exclusive to the Feces

From the 12 samples taken from the six ordinary rats, we obtained 1050 kinds of OTUs (range, 198–874), 490.00 ± 268.20 from the ileum and 430.50 ± 31.97 from feces. We compared species in and between ileal and fecal samples. Large proportions of species were shared between ileal and fecal samples (Figure 2A,B). However, a great many species remained exclusive to the ileum, such as 36.84% of phyla and 58.31% of genera (Figure 2C). That is, a considerable number of ileal bacteria were not detected in feces. After annotation, we found that the phyla Chloroflexi, Fusobacteria, Latescibacteria, Nitrospirae, Planctomycetes, Rokubacteria, and unassigned bacteria were specific to the ileum, with the OTUs ranging from 3 to 532. At the genus level, 193 bacteria were ileum exclusive. All had low abundance, with 61.82 ± 113.95 OTUs (SEM: 8.22) and a prevalence of 17–100% (Appendix A, Appendix A), except for *Acetobacter* (Proteobacteria), with 1798 OTUs.

### 3.2. Shared Bacteria Were Composed of Distinct Floral Structures in the Ileum and Feces

Shared bacteria, which can be defined as those transferred from the ileum to the colon along with digesta, constituted the main body of intestinal flora by virtue of their high abundance. The top 10 phyla—namely Firmicutes, Bacteroidetes, Actinobacteria, Proteobacteria, Verrucomicrobia, Spirochaetes, Patescibacteria, Epsilonbacteraeota, Acidobacteria, and Tenericutes (OTUs range, 36–152,356)—as well as Cyanobacteria and Gemmatimonadetes (OTUs < 1000), were the phyla of bacteria shared between the ileal and fecal flora. They showed obviously different ratios of abundance. High ratios of Patescibacteria, Verrucomicrobia, Spirochaetes, Bacteroidetes, and Tenericutes were in feces, while Firmicutes, Proteobacteria, Acidobacteria, Epsilonbacteraeota, and Actinobacteria thrived more in ileal flora (Figure 3A). At the genus level, *Romboutsia* (138,023 OTUs), *Turicibacter* (63,554 OTUs), and *Rothia* (12,466 OTUs) dominated the constitution of ileal bacteria (Figure 3B), with levels 6–14 times higher than in feces. Moreover, *Streptococcus* (OTUs in feces: 136; in ileum: 4771), *Lactobacillus* (OTUs in feces: 23; in ileum: 4598), and *Escherichia-Shigella* (OTUs in feces: 29; in ileum: 3251) were also dominant in the ileum (35–200 times higher than in feces; Figure 3C). An additional 47 shared ileum-dominant bacteria had low abundances (<2000 OTUs ileum; Appendix A). The dominant microbiota also varied across individuals (Figure 3C).

### 3.3. Differing Diversity, Biomarkers, and Functions in Ileal and Fecal Bacteria

We further studied the microbiotal diversity and biomarkers of ileal and fecal flora. Results showed that the species alpha diversity of ileal and fecal microbiota were identical (*p* > 0.05; Appendix A), mostly due to the large variance of ileal flora. According to PCA and PCoA analyses by the binary Jaccard method, the β-diversity between samples was significantly higher in ileal than in fecal flora (PERMANOVA, *r^2^* = 0.345, *p* = 0.008; Figure 4A,B). Moreover, according to the unweighted pair group method with arithmetic mean (UPGMA) and heatmap analyses, the distance between in-group samples was shorter than that between paired ileal–fecal samples. That is to say, the microbiota were more similar across individuals than across body sites from the same individual (Figure 4C,D). Linear discriminant analysis with effect size (LEfSe) analysis showed that ileal and fecal bacteria had distinct biomarkers (Figure 4E). Taken together, these results indicated that ileal and fecal flora were different despite the bulk of dominant bacteria being continuously transferred from the ileum to the colon along with dietary residues.

Based on the β-diversity analysis, bacterial prevalence and abundance varied across individuals and was host specific. In regular rats, only 69 of 337 ileal bacteria and 75 of 337 fecal bacteria were 100% prevalent. All less-prevalent bacteria were also less dominant, mostly with OTUs < 300. Among the 100% prevalent bacteria, only a few were highly dominant, making up the top 10 dominant bacteria of the ileal or fecal group. Meanwhile, the dominant bacteria varied across individuals. *Romboutsia* was dominant in ileal samples from all six individual rats, whereas *Turicibacter* and *Rothia* were dominant in three individuals only. In fecal microbiota, *uncultured bacterium f Muribaculaceae* was dominant in all six individuals, *Lachnospiraceae NK4A136 group* in four, and *Treponema* 2 in only two.

Functionally, according to Tax4Fun (http://tax4fun.gobics.de/, access on 28 February 2020) predictions, ileal bacteria displayed higher levels of metabolism (74.52 ± 0.71% vs. 69.28 ± 1.07%) and organismal systems (1.53 ± 0.14% vs. 1.16 ± 0.10%) but a lower level of environmental information processing (10.69 ± 0.70% vs. 15.54 ± 1.02%) than fecal flora (*p* < 0.05) (Appendix A).

### 3.4. Ileal and Fecal Microbiota Under High-Fat Diet Intervention

To validate the structures and features of ileal flora and clarify dietary diversity’s possible persistent influence on them, we established another group of rats for which the intervention was a HFD. After being fed the HFD for 60 days, the animals were then switched to chow for another 60 days (n = 10). We collected the rats’ intestinal contents and subjected them to 16S rRNA gene V3–V4 region sequencing and analysis. Results showed obvious differences between the ileal and fecal bacteria mentioned earlier. The ileum-exclusive bacteria constituted 26.32% in phyla and 53.57% in genera (Figure 5A). Latescibacteria, Nitrospirae, Planctomycetes, Gemmatimonadetes, and unassigned bacteria were specific to the ileum (OTU range, 16–503). In addition, 180 genera of bacteria were ileum exclusive, with OTUs ranging from 6 to 725 (96.48 ± 115.04), except for *uncultured bacterium c Subgroup 6* (*Acidobacteria*) with 1483 OTUs (Appendix A). *Romboutsia* (200,712 OTUs), *Rothia* (69,437 OTUs), *Turicibacter* (28,073 OTUs), 5–24 times higher than in feces, *Streptococcus* (23,487 OTUs), *Helicobacter* (13,346 OTUs), and *Candidatus Arthromitus* (22,921 OTUs), 44–997 times higher than in feces, dominated the constitution of ileal bacteria. Another 53 shared ileum-dominant bacteria had low abundances (<5056 in SI; Appendix A). Moreover, they showed the same functional differences as those of chow-fed rats. However, the HFD did exert a lasting influence on the dominant categories and α-diversity of ileal bacteria, which might be biomarkers of specific-diet ingestion. The richness, diversity, and evenness of ileum flora were all elevated (Appendix A). The β-diversity of the fecal microbiome was significantly lower than that of the ileal microbiome (PERMANOVA, *r^2^* = 0.316, *p* = 0.001). The fecal bacteria also showed a higher similarity and a more consistent abundance than those between paired ileal–fecal samples (Figure 5B). A cladogram analysis showed that ileal and fecal bacteria had distinct biomarkers (Figure 5C).

### 3.5. Ileal Microbiota Under Specific-Pathogen–Free Conditions

We also investigated the features of the ileal bacteria of clean rats in germ-free conditions. Their ileal and fecal contents produced a total of 1254 kinds of OTU (ileum: 1013 [393.67 ± 77.19]; feces: 1005 [482.83 ± 37.38]; *p* > 0.05). The SPF environment remarkably influenced the intestinal bacterial structure. The variability of ileal bacteria and exclusive bacteria was nearly diminished. Only Dependentiae (1 OTU) and 89 bacteria genera were ileum exclusive, with very low abundance (OTUs: 1.78 ± 1.16 [range, 1–6]). *Lactobacillus*, *Romboutsia*, and *Candidatus Arthromitus* were dominant (Figure 6A). The abundances of less-dominant shared bacteria were also decreased (Appendix A). The Shannon and Simpson indices were lower for ileal than for fecal flora (*p* < 0.05; Appendix A). The differences between ileal and fecal bacteria, however, were remarkable (Figure 6B,C).

Functionally, ileal bacteria also had a higher level of genetic information and environmental information processing in addition to fewer cellular processes and lower metabolic abundance than fecal flora (Figure 7A). In the bugbase phenotype prediction (https://bugbase.cs.umn.edu/, access on 25 November 2020), ileal bacteria had advantages in regard to mobile genetic elements (MGEs), Gram positivity, and facultative anaerobism, whereas anaerobism, Gram negativity, and potentially pathogenic phenotypes were significantly lower than fecal flora (Figure 7B). In the Functional Annotation of Prokaryotic Taxa (FAPROTAXC) [27] analysis, the ileal bacteria also showed lower nitrate reduction, nitrification, xylanolysis, and aerobic ammonia oxidation than fecal flora (Figure 7C).

## 4. Discussion

Intestinal microbiota have increasingly become a popular research topic in recent years. They are important symbionts of the gut and are involved in physiological or pathological processes [28,29,30]. In humans, research on gut microbiota has mainly been performed using patients’ feces [15,17]. However, whether fecal bacteria can represent all microbial populations throughout the gut remains unclear. Many studies have deduced distinctions between fecal microorganisms and other components of intestinal microbiota according to functional heterogeneity [31,32,33] and the complicated biochemical environment (e.g., oxygen, pH, mucous thickness, antimicrobials, bile acids, transit time) [1]. Mentula et al. compared the microbiota of the jejunal fluid and fecal samples of 22 beagles by means of organismal culture and proved the inability of fecal samples to represent the microbiota of the upper gut. Approximately 25% of jejunal bacteria were not detectable in fecal samples, nor were 45% of fecal bacteria detectable in the jejunum [32]. Recently, a study from 14 different locations in the GI tracts of three genetically homogenous sibling pigs evaluated the validity of using feces as a proxy for the whole gut microbiome; a comparative analysis of the microbiome and metabolites in feces and in the whole GI tract proved that the fecal microbiome cannot sufficiently represent the whole gut microbiome [34]. The studies of microbial community comparisons between the small intestine and feces are summarized in Table 1.

In this study, we found that 58.30% and 53.57% of ileal bacteria were exclusive to feces and that 2.82% and 0% of fecal bacteria were exclusive to the ileum in regular and HFD rats under conventional conditions, respectively; in the SPF environment, those chow values were, respectively, 20.51% vs. 16.87%. Exclusive bacteria were all in very low abundance. We analyzed the microecological characteristics of intestinal bacteria. Large proportions of both ileal and fecal bacteria were in low abundance, and only very small proportions were in extremely high abundance, which included the top 10 bacteria that contributed 70–80% of the abundance. Accordingly, in a certain segment of the gut, bacteria could be classified into the following two groups: very few high-abundance bacteria (dominant) and a great many low-abundance bacteria (less dominant) (Appendix A). The majority of less-dominant bacteria may or may not transition with gut content (shared or exclusive), maintain their abundance at a relatively constant level, and comprise the majority of the segment’s microbial ecosystem. Dominant bacteria make up a very small proportion of bacteria (3/80 of the shared bacteria in regular rats, 8/80 in HFD rats, and 3/80 in SPF rats), with OTUs > 10,000. *Romboutsia* was the leading bacterium, with OTUs > 100,000. Dominant bacteria were distinct between the ileum and feces. *Escherichia-Shigella*, *Lactobacillus*, *Romboutsia*, *Rothia*, *Streptococcus*, and *Turicibacter* were dominant in the ileum but not in feces (reduced by 7–200 times), whereas the reverse was true of Lachnospiraceae NK4A136 group, *uncultured bacterium f Muribaculaceae*, *Treponema 2*, *uncultured bacterium f Lachnospiraceae*, *Akkermansia*, *Candidatus Saccharimonas*, and *Ruminococcaceae UCG-013* (Appendix A) (increased by 5–63 times). Meanwhile, bacteria also varied across individuals (Appendix A). Of ileal bacteria, only *Romboutsia* was dominant in all six individuals, while *Turicibacter* and *Rothia* were dominant in only three. In fecal microbiota, dominant *uncultured bacterium f Muribaculaceae* was prevalent while dominant *Lachnospiraceae NK4A136 group* and *Treponema 2* were less prevalent. The less prevalent bacteria and various dominant bacteria determined the microbiotal biomarker of unique individuals. These results suggested that analysis of fecal microbiota was not sufficient for determining the bacterial dominance of ileal microbiota; the results also indicated that fecal bacteria were limited in their ability to represent ileal bacteria and vice versa.

We also determined the influence of diet and environmental conditions on the gut microbiome. We showed that the HFD-associated bacteria exhibited durability and stability, maintaining dietary memory in the gut. The SPF environment condition decreased the variability of ileal bacteria; thus, the exclusive bacteria of the ileal flora were associated with the external environment. However, unlike varied diets, environmental conditions did not change the differences in the microecological structure and function between the ileal and fecal flora, suggesting that the enteral nutritional environment is the key factor for determining the segmental distribution of gut microbiota [34].

Humans have similar gut physiological processes to rats in regard to content degradation [35], feces transition from the small intestine to the large intestine, and staging. The results of the segmental distribution of the gut microbiome in this study suggested that the feces collected clinically for research or medications may only represent part of the microbiome information of the small intestine. The dominant bacteria in feces may fail to represent that of the small intestine. The less dominant bacteria have an advantage in species diversity that constitutes the gut microecology. The diverse diets and living conditions of humans may decrease the representativeness of feces in regard to the small intestine microbial community or the prevalence across populations. One limitation of this study is the lack of a microbiome comparison between the human ileum and feces. Moreover, a detailed location and tracing experiment should be conducted to compare the microbiome profile of the ileum content with its own feces in subsequent time points.

The segmental and less prevalent distribution of intestinal microbiota may provide valuable significance for the microbiota disease mechanisms of human beings. First, many diseases have been reported to be associated with changes in gut bacteria. Gut microbes have become a key mechanism of diseases. However, caution should be taken when concluding that there are associations between small intestinal absorption, metabolism, and gut leakage from feces. Moreover, dominance and prevalence should be sufficiently considered when aiming to identify disease-relevant specific phyla or genera in bacteria. Conducting individualized, pre- or post-disease self-controlled research is an important experiment consideration. Second, bacteria-based treatment has been a useful strategy for pathogenesis study and clinical therapy, such as fecal microbiota transplantation (FMT) [37,38]. It is important to select segment-specific bacteria for segment-specific intestinal diseases for precision medicine [35,36], just like one should induce a gastroduodenal peptic ulcer with *Helicobacter pylori* [39] or treat obesity with *Fusimonas intestini* [40]. Third, as far as gut diseases are concerned, they also possess more segmental and less prevalent features than gut microbes. The less abundant exclusive bacteria may be of greater value in searching for biomarkers specific to gut disease than the dominant bacteria, such as *Helicobacter pylori* to gastric cancer. Fourth, diet, feeding environment, age, geography, and enteral microenvironment (oxygen, pH, mucous thickness, antimicrobials, bile acids, transit time) have profound influences on gut microbes [1,34,41]. Thus, the question of whether gut microbiota features are the reason or the consequence of disease needs further investigation.

## 5. Conclusions

Collectively, we proved that the ileal contents and feces have similar microecological characteristics, including dominant and non-dominant bacteria. The different nutrient environments in different individuals and their different segmentation have a significant impact on the growth and abundance of intestinal microbiota. Different sites and individuals have different dominant, less dominant, and exclusive bacteria with different alpha- and beta-diversities. Feces can only represent about 40% of ileal bacteria with low abundance. There is a greater risk of using fecal flora to represent the entire intestinal flora. For studies of low-abundance disease-related bacterial populations, it is important to select specific samples and avoid using feces instead. Additionally, it is necessary to conduct individualized pre- and post-disease controlled studies.

## Figures and Tables

**Figure 1 microorganisms-12-01764-f001:**
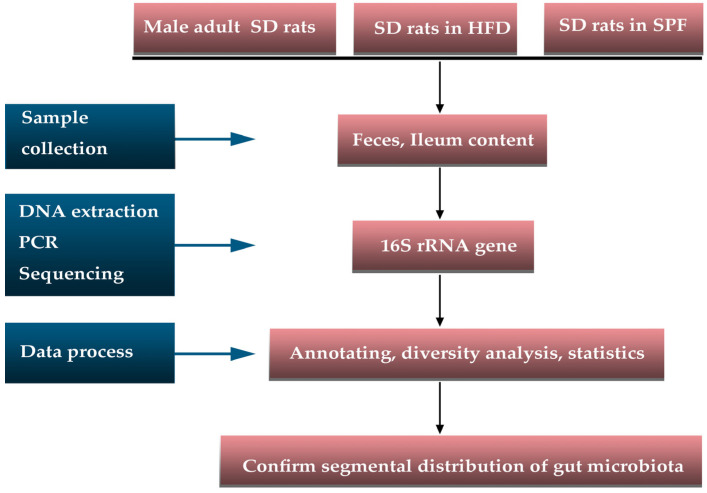
Flow chart of the experiment. The 16S rRNA gene sequencing was performed on the feces and ileum content of male adult SD rats (n = 6), SD rats with a high-fat diet (HFD) (*n* = 10), and SD rats in a specific pathogen free (SPF) environment (*n* = 6) were used to confirm the segmental distribution of gut microbiota. The black arrow indicated the experimental process, and the blue arrow indicated the scheme description.

**Figure 2 microorganisms-12-01764-f002:**
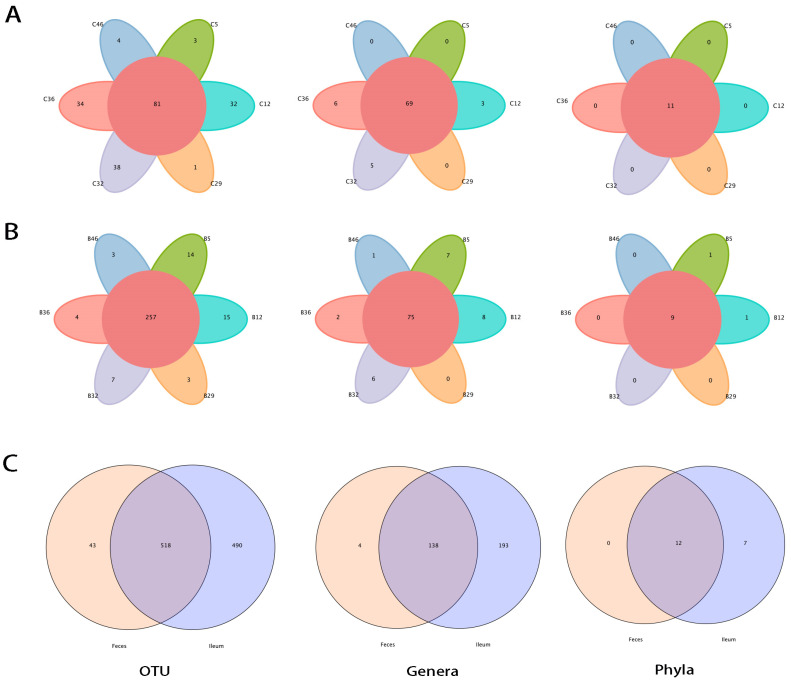
OTU and species (genera and phyla) relationships in and between ileal and fecal samples shown as Venn diagrams. Most OTUs and species were shared among individuals, but the ileum (**A**) shared fewer OTUs than feces (**B**). (**C**) Compared with fecal samples, large proportions of ileal OTUs and species were ileum exclusive, as follows: OTU: feces exclusive, 195.8 ± 97.7; ileum exclusive, 255.3 ±233.8; shared 234.7 ± 70.0; Genera: feces exclusive, 26.0 ± 13.4; ileum exclusive, 112.3 ± 85.2; shared 80.0 ±14.5; Phyla: feces exclusive, 0.3 ± 0.5; ileum exclusive, 5.3 ± 3.2; shared 9.8 ± 1.0, n = 6. R v3.1.1, VennDiagram-v1.6.9.

**Figure 3 microorganisms-12-01764-f003:**
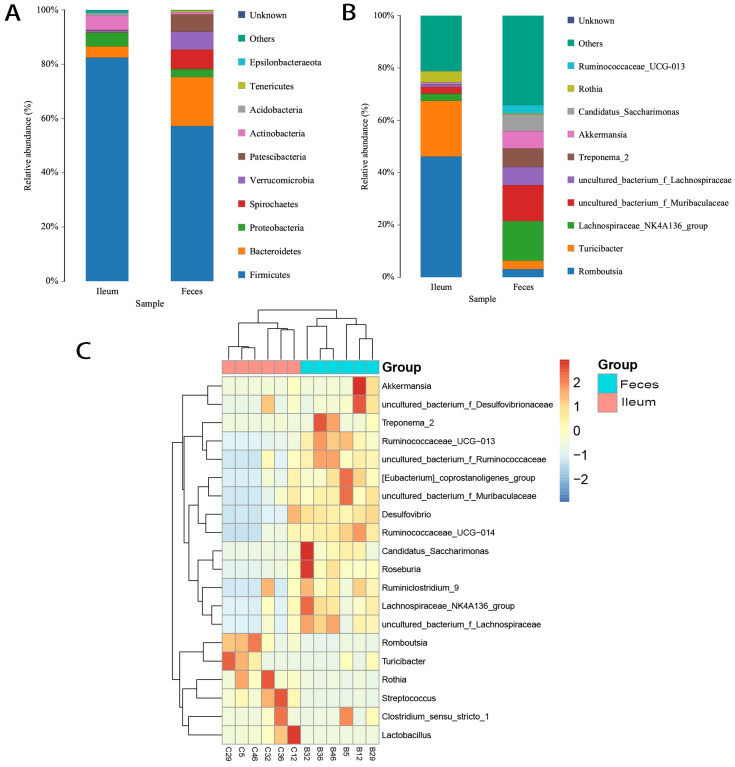
Top 10 abundant bacteria in feces and the ileum and top 20 dominant bacteria in individuals. Bacteria exhibited notably different abundances and dominance levels between feces and the ileum. (**A**) Phyla: Firmicutes thrived in the ileum while Bacteroidetes thrived in feces. (**B**) Genera: *Romboutsia*, *Turicibacter*, and *Rothia* thrived in the ileum and *Lachnospiraceae_NK4A136_group*, *uncultured_bacterium_f_Muribaculaceae*, *Akkermansia*, *uncultured_bacterium_f_Lachnospiraceae*, *Candidatus_Saccharimonas*, and *Treponema_2* thrived in feces. Python2, matplotlib-v1.5.1. (**C**) The dominant microbiota varied across segments of the gut and individuals. Heatmap showing the top 20 abundant bacteria in feces and ileum in individuals at the genus level. The distance was calculated by the Euclidean method; abundance ratio > 1%, R v3.1.1, pheatmap, v1.0.2.

**Figure 4 microorganisms-12-01764-f004:**
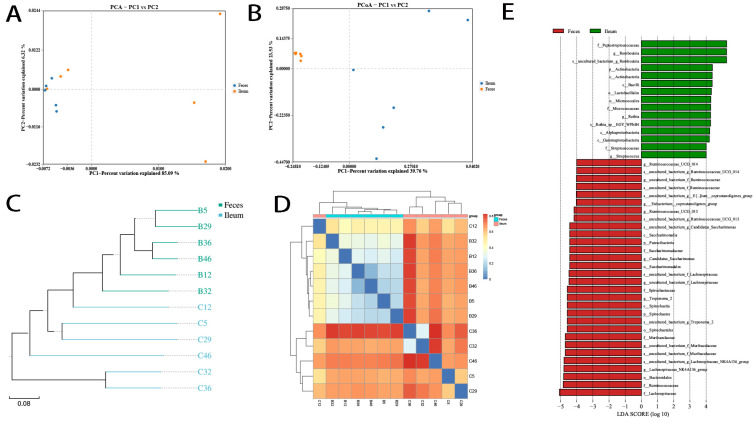
The beta-diversity, sample distances, and biomarkers of ileal and fecal bacteria. Ileal bacteria showed higher β-diversity and sample distances than fecal bacteria and shared different biomarkers. (**A**) The PCA of microbial communities from the ileum and feces based on the weighted UniFrac distances of the detected OTUs (python2, sklearn 0.17.1). (**B**) The PCoA of microbial communities from the ileum and feces based on the weighted UniFrac distance of detected OTUs (qiime 1.8.0, principal_coordinates.py). (**C**) UPGMA was used to construct an inter-sample tree structure to reflect the differences in community composition between the ileum and feces samples (python2, ete3, v3.0.0b35). (**D**) A distance heatmap showing the inter-sample matrix (R software v3.1.1, pheatmap, v1.0.2). (**E**) LeFSe biomarkers (linear discriminant analysis [LDA] threshold = 4.0) (python2, lefse). Distances were calculated using the binary Jaccard method.

**Figure 5 microorganisms-12-01764-f005:**
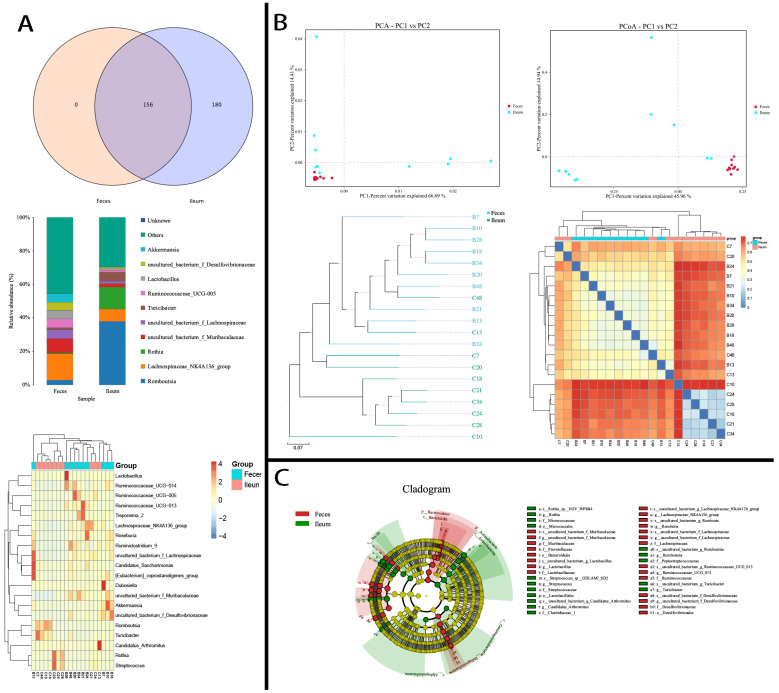
Ileal and fecal flora and diversity under HFD intervention. (**A**) Genera of ileum-exclusive bacteria. Top, Venn diagram (R v3.1.1, VennDiagram-v1.6.9); middle, bar plot showing the top 10 dominant bacteria between the feces and ileum (python2, matplotlib-v1.5.1); bottom, heatmap showing the different distributions of dominant bacteria (R v3.1.1, pheatmap, v1.0.2). (**B**) An analysis of distance between samples showing that the sample distance in fecal bacteria was shorter than that in ileal bacteria. Top, PCA (python2, sklearn 0.17.1) and PCoA (qiime 1.8.0, principal_coordinates.py) analyses; left bottom, UPGMA plot (python2, ete3, v3.0.0b35); right bottom, heatmap of sample distances (R software v3.1.1, pheatmap, v1.0.2). (**C**) Cladogram showing different biomarkers between fecal and ileal bacteria (python2, lefse).

**Figure 6 microorganisms-12-01764-f006:**
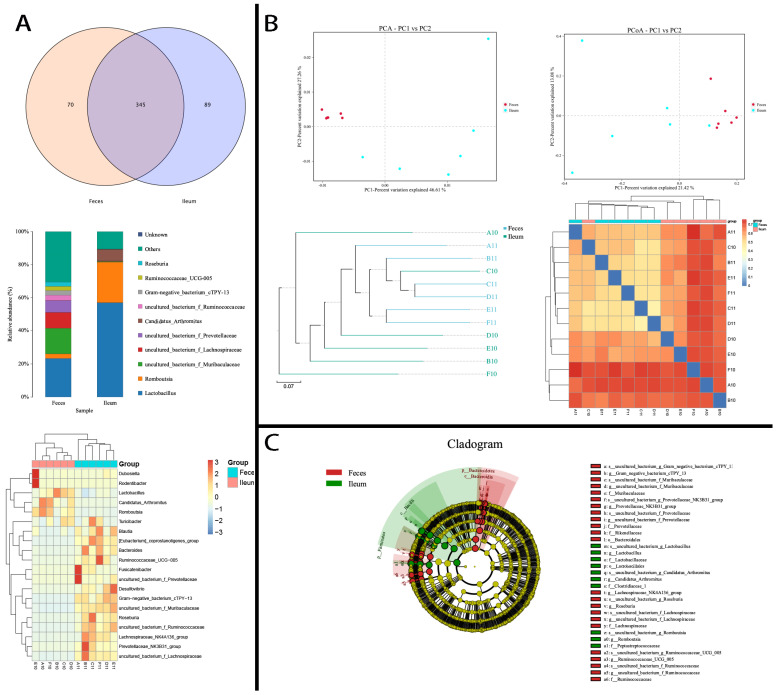
Ileal and fecal flora under SPF conditions. (**A**) The genera of ileum-exclusive bacteria. Top, Venn diagram (R v3.1.1, VennDiagram-v1.6.9); middle, bar plot showing the top 10 dominant bacteria between the feces and ileum (python2, matplotlib-v1.5.1); bottom, heatmap showing the different distributions of dominant bacteria (R v3.1.1, pheatmap, v1.0.2). (**B**) An analysis of distance between the samples showing that the sample distance in fecal bacteria was shorter than that in ileal bacteria. Top, PCA (python2, sklearn 0.17.1) and PCoA (qiime 1.8.0, principal_coordinates.py) analyses; left bottom, UPGMA plot (python2, ete3, v3.0.0b35); right bottom, a heatmap of the sample distances (R software v3.1.1, pheatmap, v1.0.2). (**C**) A cladogram showing the different biomarkers between fecal and ileal bacteria (python2, lefse).

**Figure 7 microorganisms-12-01764-f007:**
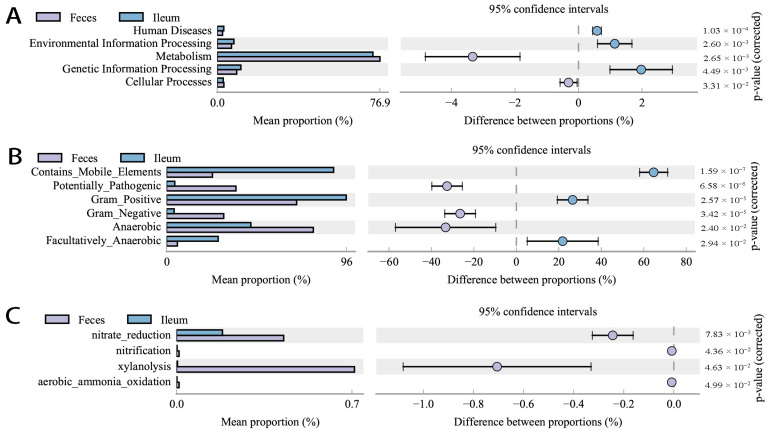
Functional prediction of fecal and ileal bacteria under SPF conditions. (**A**) PICRUSt 2 analysis (picrust2, 2.3.0). (**B**) Bugbase phenotype prediction (bugbase, 0.1.0). (**C**) FAPROTAX function analysis (FAPROTA, 1.2.6).

**Table 1 microorganisms-12-01764-t001:** The main studies comparing the microbial communities of the small intestine and feces.

Subject	Sex	Sampling	Analysis Methods	Sampling Locations	Relationship	Year of Reported
Beagle dogs	M	Permanent jejunal fistula	Culture	Jejunum fluid, feces	Fecal samples were unable to represent the microbiota present in the upper gut.	2005 [32]
Human	M/F	Esophagogastro- duodenoscopy	16S rRNA gene sequencing	Stomach, duodenum, jejunum, feces	The small bowel microbiome was markedly different from that in the stool and varied between segments.	2020 [17]
Mongolian horses	M/F	Postmortem	16S rRNA gene sequencing	Stomach, foregut (jejunum, ileum), hindgut (cecum, colon)	The microbial community structures were significantly different among the stomach, foregut, and hindgut.	2020 [18]
Pig	M	Postmortem	16S rRNA gene sequencing	Whole gut	The fecal microbiome was insufficient to represent the whole gut microbiome.	2023 [34]
Human	M/F	Sampling capsule	16S rRNA gene sequencing	Small intestinal fluid, feces	Analysis of stool provided neither a complete nor an accurate representation of the longitudinal and temporal variability of the microbiota composition, virus activity, host proteome or bile acid contents within the intestines	2023 [35]
Beagle dogs	M/F	SIMBA™ capsule	16S rRNA gene sequencing	Small intestinal fluid, feces	A statistically significant difference in the microbial composition of capsules and feces was found.	2023 [36]

M/F: Male/Female; SIMBA™: small intestinal microbiome aspiration.

## Data Availability

The raw datasets generated during the current study are available in the NCBI repository (https://www.ncbi.nlm.nih.gov/, access on 25 March 2022 and 24 April 2022), BioProject: PRJNA820028 and PRJNA831335.

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
