# Peer review of "Structural and Functional Differences in Small Intestinal and Fecal Microbiota: 16S rRNA Gene Investigation in Rats"

_microorganisms, 2024, doi:10.3390/microorganisms12091764_

Round 1
Reviewer 1 Report
Comments and Suggestions for Authors
Dear Authors,
you have prepared a valuable article MS.
The main results: Large proportions of ileal flora were exclusive to the feces; Shared bacteria were composed of distinct floral structures in the ileum and feces; Differing diversity, biomarkers, and functions in ileal and fecal bacteria were found; Ileal and fecal microbiota under high-fat diet intervention differs from normal diet; ileal bacteria also had a higher level of genetic-information processing
and fewer cellular processes than fecal flora; The SPF environment remarkably influenced intestinal bacterial structure. Studying ileal microbiota under specific-pathogen–free conditions was found that the variability of ileal bacteria and exclusive bacteria was nearly diminished. Dominant bacteria were distinct between the ileum and feces. It should be used italics font for Bacteria species names. It is necessary to add Conclusion part after 4.Discussion.
Please use italics font for Bacteria species names. Add Conclusion part after 4.Discussion.
Author Response
|
Response to Reviewer 1 Comments |
|||
|
1. Summary |
|
|
|
|
Thank you very much for taking the time to review this manuscript. Please find the detailed responses below and the corresponding revisions and corrections highlighted or marked with tracked changes in the resubmitted files. |
|||
|
2. Questions for General Evaluation |
Reviewer’s Evaluation |
Response and Revisions |
|
|
Does the introduction provide sufficient background and include all relevant references? |
Yes |
Thank you for the affirmation. |
|
|
Are all the cited references relevant to the research? |
|
There is no comment. |
|
|
Is the research design appropriate? |
Yes |
Thank you for the affirmation. |
|
|
Are the methods adequately described? |
Yes |
Thank you for the affirmation. |
|
|
Are the results clearly presented? |
Yes |
Thank you for the affirmation. |
|
|
Are the conclusions supported by the results? |
Not applicable |
We have added a Conclusion Section after Section 4, Discussion. |
|
|
3. Point-by-point Responses to Comments and Suggestions for Authors |
|||
|
Comment 1: You have prepared a valuable article MS. |
|||
|
Response 1: Thank you for your affirmation and careful reading of our manuscript. |
|||
|
Comment 2: |
|||
|
It should be used italics font for Bacteria species names. Response 2: We agree and have, accordingly, formatted the bacteria species names in genus level with italics. The changes are tracked in the revised manuscript. |
|||
|
Comments 3: It is necessary to add Conclusion part after 4.Discussion. Response 3: We have added a Conclusion Section after Section 4, Discussion. The relevant information can be found on page 14, paragraph 4. |
|||
|
4. Response to Comments on the Quality of English Language |
|||
|
Point 1: English language fine. No issues detected. |
|||
|
Response 1: Thank you for your affirmation and careful reading of our manuscript.
|
|||
|
5. Additional clarifications |
|||
|
None. |
|||

Reviewer 2 Report
Comments and Suggestions for Authors
The manuscript "Structural and Functional Differences in Small Intestinal and Fecal Microbiota: 16S rRNA Gene Investigation in Rats" provides interesting information on fecal microbiota and gene investigation. Suggestions:
1. The introduction is well done.
2. In the materials and methods section, include a flow chart.
3. The results are well presented, but specify the software used under each image.
4. In the discussion section, present the practical importance in various human pathologies: I recommend the following articles to enhance the manuscript's value:
-https://doi.org/10.3390/biomedicines11112930
-https://doi.org/10.3390/diagnostics14090861
-https://doi.org/10.3390/biomedicines11041016
5. In the discussion section, add a table with the main studies from the literature that have studied the same topic and their findings.
6. Add a conclusion section.
7. Use the MDPI style for the bibliography.
Author Response
|
Response to Reviewer 2 Comments |
||
|
1. Summary |
|
|
|
Thank you very much for taking the time to review this manuscript. Please find the detailed responses below and the corresponding revisions and corrections highlighted or marked with tracked changes in the resubmitted files.
|
||
|
2. Questions for General Evaluation |
Reviewer’s Evaluation |
Response and Revisions |
|
Does the introduction provide sufficient background and include all relevant references? |
Can be improved |
We have added some key information according to the comments. |
|
Are all the cited references relevant to the research? |
|
There is no comment. |
|
Is the research design appropriate? |
Can be improved |
We have added some key information according to the comments. |
|
Are the methods adequately described? |
Must be improved |
We have added some key information according to the comments. |
|
Are the results clearly presented? |
Can be improved |
We have added some key information according to the comments. |
|
Are the conclusions supported by the results? |
Must be improved |
We have added some key information according to the comments. |
|
3. Point-by-point Responses to Comments and Suggestions for Authors |
||
|
Comment 1: The introduction is well done. Response 1: Thank you for your affirmation and careful reading of our manuscript. Comment 2: In the materials and methods section, include a flow chart. Response 2: Thank you for suggesting this. We agree with this comment. Therefore, we have added a flow chart in the Materials and Methods Section. The relevant information can be found on page 3, Figure 1. Comment 3: The results are well presented, but specify the software used under each image. Response 3: Thank you for pointing this out. We agree with this comment. Therefore, we have added the software used under each image. The relevant information can be found in the legends of Figure 2 to Figure 7. Comment 4: In the discussion section, present the practical importance in various human pathologies: I recommend the following articles to enhance the manuscript's value: -https://doi.org/10.3390/biomedicines11112930 Boicean A, Birlutiu V, Ichim C, Brusnic O, Onișor DM. Fecal Microbiota Transplantation in Liver Cirrhosis. Biomedicines. 2023 Oct 30;11(11):2930. doi: 10.3390/biomedicines11112930. PMID: 38001930; PMCID: PMC10668969. -https://doi.org/10.3390/diagnostics14090861 Boicean A, Ichim C, Todor SB, Anderco P, Popa ML. The Importance of Microbiota and Fecal Microbiota Transplantation in Pancreatic Disorders. Diagnostics (Basel). 2024 Apr 23;14(9):861. doi: 10.3390/diagnostics14090861. PMID: 38732276; PMCID: PMC11082979. -https://doi.org/10.3390/biomedicines11041016 Boicean A, Birlutiu V, Ichim C, Anderco P, Birsan S. Fecal Microbiota Transplantation in Inflammatory Bowel Disease. Biomedicines. 2023; 11(4):1016. https://doi.org/10.3390/biomedicines11041016 Response 4: Thank you for pointing this out. We agree with this comment. Therefore, we have presented the practical importance regarding human diseases. The relevant information can be found on page 14, paragraph 3. The references the reviewer recommended are very useful. We added two references mentioned above as references 36 and 37. Comment 5: In the discussion section, add a table with the main studies from the literature that have studied the same topic and their findings. Response 5: Thank you for suggesting this. We agree with this comment. We have added a table summarizing the main studies and findings on this topic. The relevant information can be found on page 11 in Table 1. Comment 6: Add a conclusion section. Response 6: We have added a Conclusion Section after Section 4., Discussion. The relevant information can be found on page 14, paragraph 4. Comment 7: Use the MDPI style for the bibliography. Response 7: Thank you for pointing this out. We agree with this comment. We will update the bibliography style in the final version. |
||
- Response to Comments on the Quality of English Language
Point 1: English language fine. No issues detected.
Response 1: Thank you for your affirmation.
- Additional clarifications
None.

Reviewer 3 Report
Comments and Suggestions for Authors
The authors of this article show that the fecal microbiome and the ileal microbiome differ in terms of the microbial species diversity and dominance thereby highlighting that the fecal microbiome can not faithfully represent the microbiome of the gastrointestinal tract.
Following are my comments:
1. The figure panels should be marked all through the result section to improve readability.
2. Recent studies have shown differences in the microbial (fecal) diversity when sampled at different times during the day. Would sampling feces at different times during the day better represent the microbial diversity of the ileum?
Comments on the Quality of English LanguageEnglish is satisfactory and needs minor edits.
Author Response
|
Response to Reviewer 3 Comments
|
||
|
1. Summary |
|
|
|
Thank you very much for taking the time to review this manuscript. Please find the detailed responses below and the corresponding revisions and corrections highlighted or marked with tracked changes in the resubmitted files.
|
||
|
2. Questions for General Evaluation |
Reviewer’s Evaluation |
Response and Revisions |
|
Does the introduction provide sufficient background and include all relevant references? |
Can be improved |
We have added some key information according to the comments. |
|
Are all the cited references relevant to the research? |
|
There is no comment. |
|
Is the research design appropriate? |
Yes |
Thank you for the affirmation. |
|
Are the methods adequately described? |
Yes |
Thank you for the affirmation. |
|
Are the results clearly presented? |
Must be improved |
We have added some key information according to the comments. |
|
Are the conclusions supported by the results? |
Yes |
Thank you for the affirmation. |
|
3. Point-by-point Responses to Comments and Suggestions for Authors |
||
|
The authors of this article show that the fecal microbiome and the ileal microbiome differ in terms of the microbial species diversity and dominance thereby highlighting that the fecal microbiome can not faithfully represent the microbiome of the gastrointestinal tract. Comment 1: The figure panels should be marked all through the result section to improve readability. Response 1: Thank you for pointing this out. We agree with this comment. Therefore, we have marked the figure panels throughout the Results Section. Comment 2: Recent studies have shown differences in the microbial (fecal) diversity when sampled at different times during the day. Would sampling feces at different times during the day better represent the microbial diversity of the ileum? Response 2: Thank you for pointing this out. It is very interesting that the microbial (fecal) diversity differed when sampled at different times during the day. We considered that the temporal variation of fecal microbial communities may be associated with the diverse diet structure during the day. As for the findings of this study, the temporal misalignment during sampling on the spatially distinct microbial communities between the ileum and feces could not be ignored. Should the fecal bacterial structure be identical to its ileal stage, or should the ileal bacterial structure be identical to its ileal stage? This question demands nondamaging sampling in a living body. Recently, Shalon et al. designed a device to collect gastrointestinal contents in different segments depending on the lumen pH. They collected stool samples throughout the ingestion and recovery process of the device. Their results showed a distinct difference between the small intestine and feces, and an analysis of stools provided neither a complete nor accurate representation of the longitudinal and temporal variability of the microbiota composition within the intestines (ref. 34, Shalon D, et al. Nature. 2023;617:581-591.). This result suggested that different sampling times do not change the underlying phenomenon of the low representativeness of the ileum by feces. We have also mentioned this result as a reference, and suggested it as a limitation of this study. |
||
- Response to Comments on the Quality of English Language
Point 1: English is satisfactory and needs minor edits.
Response 1: Thank you for pointing this out. We have asked LetPub (www.letpub.com) to make further modifications.
- Additional clarifications
None.

Reviewer 4 Report
Comments and Suggestions for Authors
The manuscript is of high quality, presenting a comparative analysis of the distinct microbial compositions and functions of the ileum and feces in rats. The research design is robust, utilizing a well-defined experimental framework with three distinct groups of rats: conventional, SPF, and high-fat diet. The finding that fecal bacteria represent a limited portion of ileal bacteria, with significant differences in dominant genera, is particularly novel. So, in my view, the study has novelty and may contribute to the broader understanding of gut microbiota's structural and functional diversity.
I have only a few minor suggestions for improvement.
1) In the statistics subsection, the authors should provide more detailed descriptions of the statistical methods used, particularly clarifying the use of PCA and PCoA in the context of microbiota diversity analysis, and the methods used to compare predicted functional profiles between ileal and fecal microbiota.
2) Figures 1 and 2 could benefit from more detailed legends to enhance understanding.
3) I recommend including a brief discussion of how these findings might translate to humans.
4) The study limitations should be included in the discussion section.
5) Ensure consistent formatting of references and check for missing citations (there are 36 in the text and 37 in the list).
Comments on the Quality of English LanguageThere are a few minor grammatical errors and awkward phrasings that could be improved.
Author Response
|
Response to Reviewer 4 Comments |
|||
|
1. Summary |
|
|
|
|
Thank you very much for taking the time to review this manuscript. Please find the detailed responses below and the corresponding revisions and corrections highlighted or marked with tracked changes in the resubmitted files.
|
|||
|
2. Questions for General Evaluation |
Reviewer’s Evaluation |
Response and Revisions |
|
|
Does the introduction provide sufficient background and include all relevant references? |
Yes |
Thank you for the affirmation. |
|
|
Are all the cited references relevant to the research? |
|
There is no comment. |
|
|
Is the research design appropriate? |
Yes |
Thank you for the affirmation. |
|
|
Are the methods adequately described? |
Can be improved |
We have added some key information according to the comments. |
|
|
Are the results clearly presented? |
Yes |
Thank you for the affirmation. |
|
|
Are the conclusions supported by the results? |
Yes |
Thank you for the affirmation. |
|
|
3. Point-by-point response to Comments and Suggestions for Authors |
|||
|
Comment 1: The manuscript is of high quality, presenting a comparative analysis of the distinct microbial compositions and functions of the ileum and feces in rats. The research design is robust, utilizing a well-defined experimental framework with three distinct groups of rats: conventional, SPF, and high-fat diet. The finding that fecal bacteria represent a limited portion of ileal bacteria, with significant differences in dominant genera, is particularly novel. So, in my view, the study has novelty and may contribute to the broader understanding of gut microbiota's structural and functional diversity. |
|||
|
Response 1: Thank you for your affirmation and careful reading of our manuscript. |
|||
|
Comment 2: |
|||
|
In the statistics subsection, the authors should provide more detailed descriptions of the statistical methods used, particularly clarifying the use of PCA and PCoA in the context of microbiota diversity analysis, and the methods used to compare predicted functional profiles between ileal and fecal microbiota. Response 2: Thank you for pointing this out. We agree with this comment. We have, accordingly, revised the description of the statistical methods used. The details of the PCA and PCoA analyses and the predicted functional profile comparison between the ileal and fecal microbiota were described in detail. The modifications can be found on page 3, paragraph 5. Comment 3: Figures 1 and 2 could benefit from more detailed legends to enhance understanding. Response 3: Thank you for pointing this out. We agree with this comment. We have, accordingly, added more information in the legends. Detailed modifications can be found on page 5, Figure 2 legend; and on page 6, Figure 3 legend. Comment 4: I recommend including a brief discussion of how these findings might translate to humans. Response 4: Thank you for this suggestion. We agree with this comment. We have, accordingly, added a brief discussion of how the findings might translate to humans. The modifications can be found on page 14, paragraph 2. Comment 5: The study limitations should be included in the discussion section. Response 4: Thank you for pointing this out. We agree with this comment. We have added the limitations of this study. The modifications can be found on page 14, paragraph 2, lines 9 to 12. Comment 6: Ensure consistent formatting of references and check for missing citations (there are 36 in the text and 37 in the list). Response 6: Thank you for pointing this out. We have defined all references. Reference 37 (now 41) can be found in page 15, Acknowledgement Section. |
|||
|
4. Response to Comments on the Quality of English Language |
|||
|
Point 1: Minor editing of English language required. There are a few minor grammatical errors and awkward phrasings that could be improved. |
|||
|
Response 1: Thank you for pointing this out. We have asked LetPub (www.letpub.com) to make further modifications.
|
|||
|
5. Additional clarifications |
|||
|
None. |
|||
